# *Chelidonichthys lucerna* (Linnaeus, 1758) Population Structure in the Northeast Atlantic Inferred from Landmark-Based Body Morphometry

**DOI:** 10.3390/biology13010017

**Published:** 2023-12-27

**Authors:** Inês Ferreira, Rafael Schroeder, Estanis Mugerza, Iñaki Oyarzabal, Ian D. McCarthy, Alberto T. Correia

**Affiliations:** 1Centro Interdisciplinar de Investigação Marinha e Ambiental (CIIMAR/CIMAR), Terminal de Cruzeiros do Porto de Leixões, Avenida General Norton de Matos S/N, 4450-208 Matosinhos, Portugal; 2Faculdade de Ciência e Tecnologia da Universidade Fernando Pessoa (FCT-UFP), Praça 9 de Abril 349, 4249-004 Porto, Portugal; 3Laboratório de Estudos Marinhos Aplicados (LEMA), Escola Politécnica, Universidade do Vale do Itajaí (UNIVALI), Rua Uruguai 458, Centro, Itajaí 88302-901, Brazil; 4AZTI, Sustainable Fisheries Management, Basque Research and Technology Alliance (BRTA), Txatxarramendi Ugartea z/g, 48395 Sukarrieta, Spain; 5School of Ocean Sciences, Bangor University, Askew Street, Menai Bridge LL59 5AB, UK; 6Instituto de Ciências Biomédicas Abel Salazar (ICBAS), Universidade do Porto, Rua de Jorge Viterbo Ferreira 228, 4050-313 Porto, Portugal; 7Escola de Ciências da Vida e do Ambiente (ECVA), Universidade de Trás-os-Montes e Alto Douro (UTAD), Quinta dos Prados, 5000-801 Vila Real, Portugal

**Keywords:** Triglidae, fish stocks, natural tags, geometric morphometrics, truss network

## Abstract

**Simple Summary:**

This is the first study to investigate the spatial distribution and population structure of the tub gurnard (*Chelidonichthys lucerna*) in different regions of the northeast Atlantic. To analyze fish body shape, a morphometry-based method, which helps identify variations in fish body shape that may exist due to genetic factors or environmental adaptability, was used. This study relied on *C. lucerna* individuals captured in the following three fishing areas: Conwy Bay (United Kingdom), Biscay Bay (Spain) and Matosinhos (Portugal). The findings indicate the existence of significant regional differences in fish bodies, thus highlighting the existence of distinct fish populations in the three regions. Results also suggest that the Spanish and British populations may inhabit similar habitats, as some similarities in body shape were found. To confirm these findings, we recommend future research using a holistic approach with alternative and complimentary stock assessment tools.

**Abstract:**

The study of geometric morphometrics among stocks has proven to be a valuable tool in delineating fish spatial distributions and discriminating distinct population units. Variations in fish body morphology can be linked to genetic factors or to phenotypic adaptability in response to environmental variables. The tub gurnard (*Chelidonichthys lucerna*) is a demersal species that usually lives in the bottom of the continental shelf, being widely distributed along the northeast Atlantic, Mediterranean and Black seas. Worldwide interest in the species has increased since 2006, when ICES recognized its potential for commercial exploitation. However, despite its broad geographic occurrence, to date, research on *C. lucerna* population structure at large spatial scales is still lacking. In this paper, body geometric morphometrics, using a landmark-based truss network, was applied in order to discriminate *C. lucerna* populations caught in three different fishery grounds areas along the northeast Atlantic: Conwy Bay (United Kingdom), Biscay Bay (Spain) and Matosinhos (Portugal). The results obtained in this study revealed a high overall relocation success (95%) of samples to their original locations, thus demonstrating the existence of significant regional differences and indicating that we are dealing with different fish population units. Moreover, the data revealed a partial overlap between individuals from Spain and United Kingdom, suggesting that in geographically distant areas these populations may inhabit similar environments. However, to corroborate these findings, future works using a holistic approach with alternative and complimentary stock assessment tools (e.g., genetic and phenotypic natural tags) are highly recommended.

## 1. Introduction

Fish populations can be distributed over extensive geographical areas characterized by distinctive environmental characteristics (e.g., temperature, salinity, depth, habitat and currents), which together with the challenges related to food availability and predators can exert influence over significant demographic aspects, thereby affecting the dynamics of the populations including reproductive patterns, fecundity and lifespan [1,2,3]. In addition, other factors, such as fishing pressure, anthropic pollution, habitat destruction and climate change, can also affect the abundance and distribution of fish species and consequently fish stocks dynamics [4,5,6]. Understanding population structure is, therefore, an important component of fisheries management as it allows to effectively estimate stock-wise population abundance, determine how each stock responds to fisheries exploitation or environmental changes and the potential impacts on related and dependent species, thereby ensuring species sustainability [7,8,9].

Fish morphometric variation among stocks has been shown to be a useful tool to describe fish spatial distributions and identify different population units, as fish body morphological differences (e.g., length, width and depth) can be associated with genetic background [10,11,12] or processes of phenotypic plasticity as a response to different environmental conditions [13,14,15]. Exposure to variations in factors, such as temperature, salinity and food availability, can result in different behavioral patterns (e.g., aggregation, migration and others) and the adoption of different adaptation strategies, which could be reflected in fish morphometric features and contribute to the definition of different phenotypic stocks [16,17,18]. The truss network system is a geometric morphometrics method commonly used for stock discrimination purposes that provides information on phenotypic traits [9,19,20]. This approach is a powerful tool for the analysis of the fish contour shape and consists of covering all or most of the animal’s body with a landmark-based uniform network that allows for the measurement of a series of distances across the body form [21].

*Chelidonichthys lucerna* [22], commonly named tub gurnard, can be found in the northeast Atlantic, from Norway and the southern North Sea extending along the Atlantic shoreline of Europe around to the British Isles, and also in the Mediterranean and Black seas and the northwest coast of Africa [23]. This species is demersal and usually inhabits sand, muddy or gravel substrates of the continental shelf in depths ranging from 20 to 318 m, but it is more abundant in inshore waters up to 150 m [24,25,26]. Following a larval pelagic phase [27,28], *C. lucerna* exhibits a particular pattern of seasonal migratory movements during the juvenile and adult stages within its depth ranges throughout the year, showing a more pronounced concentration of individuals in shallower depths during spring and summer, moving progressively into deeper waters in the winter period [29,30,31]. Recently, it has been shown that *C. lucerne*, although mainly a marine fish, can occupy and migrate among habitats with diverse salinity degrees, thereby showing high environmental plasticity and adaptation [32]. In 2006, ICES classified *C. lucerna* as a potential species for commercial exploitation and has recommended that monitoring programs should be conducted to acquire information on biological parameters for stock assessment purposes [33]. Since 2010, although with a slight decrease between 2017 and 2020, worldwide fisheries landings have shown an increasing trend, reaching 4759 tons in 2021 [34]. At present, there is no minimum landing size, allowed quotas, fishing closure seasons, or other fishery regulations. According to the International Union for Conservation of Nature (IUCN), *C. lucerna* is listed as Least Concern, but it would be helpful to quantify the population trend of this species throughout the Atlantic Ocean and Mediterranean Sea [35].

Due to its broad spatial distribution and the presence of several physical and oceanographic barriers within its wide distribution range, this species could potentially consist of various distinct population units. However, to date, information about the stock structure of this species is scarce. A study, conducted using genetic and morphological analyses of fish caught in the Black Sea, Marmara, Aegean and northeastern Mediterranean coasts of Turkey, showed that only the Black Sea population is differentiated from other populations [36]. More recently, a study conducted in the Portuguese Atlantic waters using otolith shape and microchemistry fingerprints suggested that along the mainland coast the species is, although not homogeneous, apparently a single-population unit [37]. However, its population structure at the larger northeast Atlantic spatial scale is unknown.

Therefore, the present study aimed at investigating the spatial morphological variability of *C. lucerna* among three fishery grounds (British, Cantabrian and Portuguese waters) in the northeast Atlantic using a truss network approach.

## 2. Materials and Methods

### 2.1. Sampling

A total of 129 fish were collected between October 2020 and December 2021 in three different fishery grounds in the northeast Atlantic: Conwy Bay, United Kingdom (Irish Sea), Bay of Biscay, Spain (Cantabrian Sea), and Matosinhos, Portugal (northwest Portuguese waters) (Figure 1 and Table 1). All individuals were captured using bottom trawl fishing. Upon collection, or immediately after landing, fish were transported to the laboratory in isothermal containers preserved with ice for biological processing.

### 2.2. Body Morphometric Analysis

The body morphometrics of individual *C. lucerna* were analyzed using a truss network system standard protocol [21]. All individuals were measured for standard length (SL, 1 mm) (Table 1), and their left (lateral) and dorsal (upper) sides were photographed from a fixed distance using a precise scale (10 mm) with a high-quality digital camera for body morphometric analysis, following recommendations, to minimize the effects of distortion [38]. Instructions of landmark criteria and a reference image of where to place each landmark were shared among researchers to avoid image-based bias [39].

A total of 13 and 6 landmarks were defined along the body contour on the fish’s lateral and dorsal views, respectively (Figure 2). Location coordinates of homologous landmarks were processed and digitized using tpsUtil Version 1.83 [40], and tpsDig Version 2.32 [41] software and used to determine 37 linear distances (D) of the box–truss network (Table 2).

### 2.3. Statistical Analysis

The relationship between the thirty-seven morphometric distances (D1 to D37) and fish standard length (SL) was verified using One-Way Analysis of Covariance (ANCOVA). All of them presented a significant positive correlation (*p* < 0.05). Each distance was corrected to remove the size effect, and the positive allometric relationship between variables was corrected using the following transformation [42]: DT = 10^[log(D) − β [log(SL) − log(SL_mean_)], where DT is the transformed distance, D is the original distance, β is the slope of the regression of log(D) on log(SL), SL is the standard length of the individual, and SL_mean_ is the overall mean of standard length for all locations.

For the univariate statistics, the DT dataset was checked for normality (Shapiro–Wilk test) and homogeneity of variances (Levene test), but only eight DTs (DT2, DT6, DT8, DT9, DT10, DT11, DT32 and DT35) fulfilled the parametric prerequisites. For this case, One-Way Analysis of Variance (ANOVA) was used to explore the statistical differences of each morphometric distance among the three sampling locations, followed by a Tukey post hoc test if significant differences exist (*p* < 0.05). However, the majority of DTs did not fulfil the above-mentioned prerequisites, even after being log (x + 1) transformed. For them, non-parametric statistics were then used. So, One-Way ANOVA On Ranks, followed by a Dunn’s test (*p* < 0.05), if needed, was performed.

Regarding the multivariate statistics, non-parametric tests were also performed. A permutational multivariate analysis of variance (main PERMANOVA) was used to compare the DTs among locations, and when statistically significant (*p* < 0.05), it was followed by a permutational pairwise comparisons (pseudo t-statistic).

Finally, a flexible discriminant analysis (FDA) followed by a Jackknifed re-classification matrix (leave-one-out cross-validation) was used to calculate the percentage of correctly re-classified individuals into the original location. FDA first randomly split the data into training (80%) and test (20%) sets. Predictors’ parameters were estimated by subtracting the mean of the predictor and scaling by its standard deviation. Estimated parameters were used to transform the train and test sets. The correct re-classification percentage of the discriminant functions was calculated using a Jackknifed matrix for the transformed training and test sets [43].

The univariate statistical analyses were performed using SigmaPlot 11.0. Multivariate tests were performed using R 4.3.0 [44]. A statistical level of significance (α) of 0.05 was considered. Morphometric data are presented as mean ± standard error deviation.

## 3. Results

Univariate tests showed significant differences in body morphology among the locations for 36 out of 37 DTs (Table 3). Nearly one-quarter of all measurements (DT15, DT23, DT24, DT26, DT27, DT31, DT33 and DT34) presented significant differences between all three sampling locations: 46% (DT1, DT3, DT7, DT8, DT9, DT12, DT13, DT16, DT17, DT18, DT19, DT21, DT28, DT29, DT30, DT35 and DT37) differentiated Spain from the other locations, 19% (DT4, DT5, DT6, DT11, DT20, DT22 and DT32) differentiated Portugal from the other locations, and only one measurement (DT10) differentiated the United Kingdom from other locations. Only 6% of the measurements showed no significant differences between Spain and the other two sampling locations (DT14) and between Portugal and the other two locations (DT25). DT2 was the only measurement that revealed no significant differences among the three locations.

When analyzed together, the morphometric distances also presented significant differences among the three locations (PERMANOVA, *p* < 0.05; Table 3), and all pairwise tests also revealed significant differences between the locations (pseudo *t*-test, *p* < 0.05; Table 4).

The FDA conducted showed a clear discrimination for all locations (Table 5; Figure 3), with an overall high classification accuracy reported for all three sampling locations (95%). Individuals from Portugal were reclassified correctly in 98% of the cases; it was 96% for the United Kingdom and 93% for Spain. The high discrimination pattern for Portugal was mainly driven by distances DT19 and DT16, which are related with fish posterior body height. DT7 and DT18, related with fish posterior body length, were mainly responsible for the discrimination of the United Kingdom individuals, while DT12 and DT9, related with fish anterior height, were responsible for the differences in the Spanish individuals.

## 4. Discussion

The present study aimed at investigating the differences in the body shape of *C. lucerna* along the northeast Atlantic using landmark-based truss network morphometrics.

Morphometric studies on *C. lucerna* are limited and, to date, only one study has been carried out in the Black Sea, Marmara, Aegean and eastern Mediterranean Sea [36] that combined genetics with body geometric morphometrics to analyze the existent population structure within Turkish marine waters. As in the former study, the sex of the animals was not considered in this study, since it was demonstrated that there are no morphometric differences between sexes of a related species (*C. obscurus*) [45], although the potential effect of the reproductive season on the sexual dimorphism should not be excluded.

Phenotypic variation in fish morphometric characteristics provides valuable information about population units and has long been used for stock identification as morphological differences are commonly explained as a response to dissimilar environmental conditions [7,11,46].

Advances in digital imaging systems and analytical methods in the past decades have facilitated progress and diversification of morphometric techniques, expanding the potential for using morphometric analysis as a stock identification tool [9,19,46]. In this context, landmark-based truss analysis has been successfully used alone by several authors for the discrimination of fish stocks [13,14,47] or combined with other discrimination methods such as genetics [48,49,50], otolith shape [15,17,51] and otolith elemental analyses [52,53,54].

Regarding body shape, the Spanish individuals, although bigger in SL (Table 1), after size allometric correction were found to have the smallest body measurements (75% of the total recorded distances), namely in terms of head length, heights and widths, the majority of fin lengths and distances and the anterior and posterior fish heights (DT1, DT3-DT13, DT15-DT19, DT25 and DT28 and DT37). The hydrology of the Spanish sampling location (Bay of Biscay) is particularly complex due to interactions between the general oceanic circulation, topography, highly energetic tidal currents, wind-induced currents and river inputs of freshwater, mainly located on the French coast [55,56]. Other authors have reported that body, head and fins in fish are highly affected by water velocity and fish with a streamlined morphology exhibit enhanced capability to counter hydrodynamic resistance within fast-flowing water [57,58,59]. In this context, the particular oceanographic characteristics of the Bay of Biscay could have induced some morphological adaptive variations in *C. lucerna*’s body shape and explain the smaller distances within fish shape, despite the larger size of the sampled individuals.

The United Kingdom individuals, that recorded a smaller SL (Table 1), have also recorded smaller mouth sizes and caudal peduncle areas (DT2 and DT20-DT24). Conwy Bay is an inlet of the Irish Sea, which is generally characterized by large tidal energy input from the Atlantic [60]. The Bay is recognized for its unusual and varied coastal and intertidal habitats and their associated reef communities [61], which are factors that can influence *C. lucerna*’s feeding regimes and fish habitats and explain the recorded phenotypic regional differences.

The larger overall measurements (70% of the total recorded distances) were found for the Portuguese individuals, namely in terms of head length, mouth and anterior body size and peduncle area (DT1–DT9, DT11–DT13, DT15–DT17, DT20–DT24 and DT30–DT35). Fish head and mouth sizes may reflect differential habitat use, variations in feeding behaviors or the capacity to explore different ecological niches with different types of prey [62,63,64], while the lengthening of the caudal peduncle is usually associated with fish swimming ability in strong hydrodynamic environments (e.g., water currents) [65,66,67]. The Portuguese coast presents different hydrographical features influenced by the Canary and the Portuguese currents, both connected to the North Atlantic Subtropical Gyre [68] which could induce morphological variances in body shape, namely in the peduncle area. The largest distances between fins, larger 2nd dorsal fins, higher posterior body height, larger caudal fin areas and posterior body length (DT10, DT14, DT18–DT19, DT25–DT29 and DT36–DT37) were also recorded for the United Kingdom individuals. Fish body form, fin length and location are adaptations for movement that indicate differences in habitat exploitation [69], which is aligned with the distinctive and diverse environments found in the eastern Irish Sea [60,61].

Finally, the effect of the sampling period in the truss networking results cannot be disregarded in this study. At each site, samples were collected at different times of the year, which, in conjunction with the life cycle (e.g., spawning period), may have a significant impact on body shape. It is well known that for the Mediterranean Sea this species has a protracted spawning season but with peaks occurring at different sites [26,70]. However, data for the Atlantic Ocean is limited. Anyway, a previous study that was conducted in the NE Atlantic reported that females attained maturity at smaller sizes (27.7 cm vs. 29.1 cm) and younger ages (2.7 years vs. 2.8 years) compared to males [71]. This shows that basic data about the reproductive biology of *C. lucerna* are still needed in the Atlantic waters.

Another factor that can play a role in the differences observed among the three sampling locations could be attributed to distinct regional anthropogenic influences (e.g., pollution and habitat alteration) [4,6,72]. But further investigation is needed.

## 5. Conclusions

Regardless of the reason behind the regional morphological differences, our results concerning the geometric morphometrics analyses have shown significant differences among the three sampling locations, with a high overall reallocation success (95%) of individuals to the original locations. These data indicate that *C. lucerna* individuals caught in the three fishery grounds along the northeast Atlantic do not belong to a single and homogeneous population unit, despite a slight visual overlap in the FDA between the English and Spanish individuals, which suggests that fish from these locations may somewhat inhabit similar environments. Finally, the results suggest that these fisheries should be managed regionally as different population units.

However, since the regional differences found in this study regarding the sample number, size range, temporal collection window, sex ratio and age structure of the caught individuals could somewhat confound the ontogenic effects on phenotypic body variations [73], it is recommended to conduct future studies with a holistic approach using other natural tags, such as genetics, parasites fauna and otolith chemistry. In addition, studying individuals from a broader number of sampling locations would also allow for a better understanding of the northeast Atlantic population structure.

## Figures and Tables

**Figure 1 biology-13-00017-f001:**
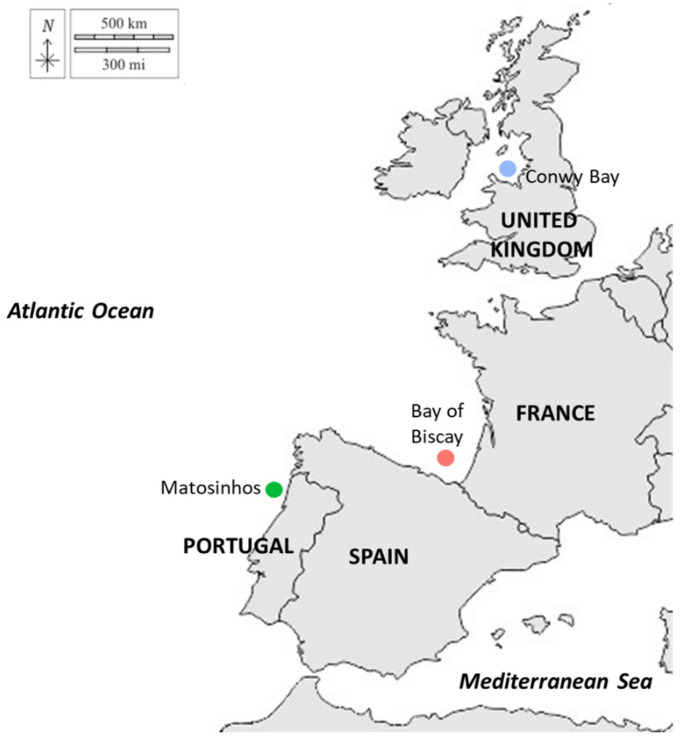
Sampling locations of *Chelidonichthys lucerna* individuals collected between October 2020 and December 2021 in the northeast Atlantic (the blue, red and green solid circles represent the eastern Irish Sea, the Cantabria Sea and the northwest Portuguese waters, respectively).

**Figure 2 biology-13-00017-f002:**
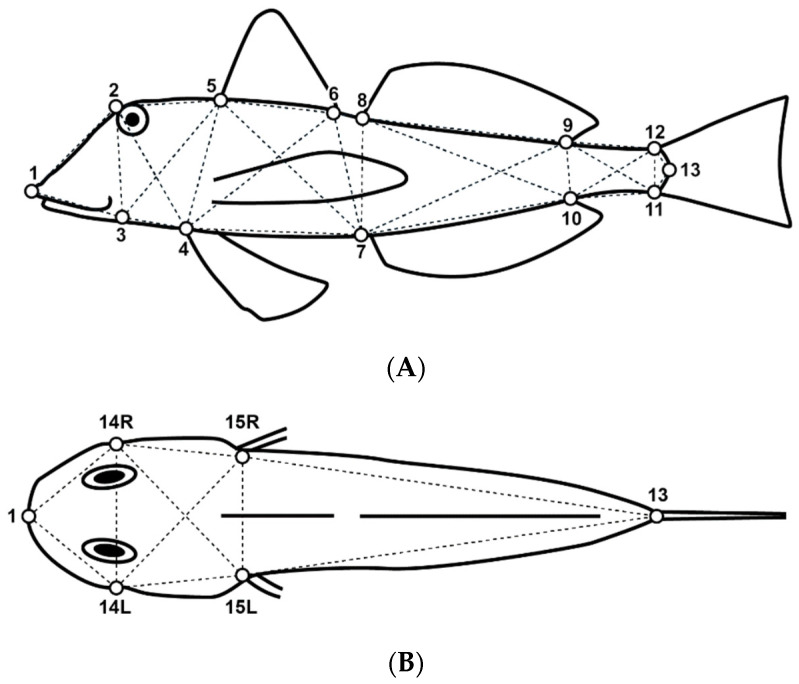
Illustration of a *Chelidonichthys lucerna* specimen showing the selected landmarks for the lateral (**A**) and dorsal (**B**) body views. See Table 2 for further details.

**Figure 3 biology-13-00017-f003:**
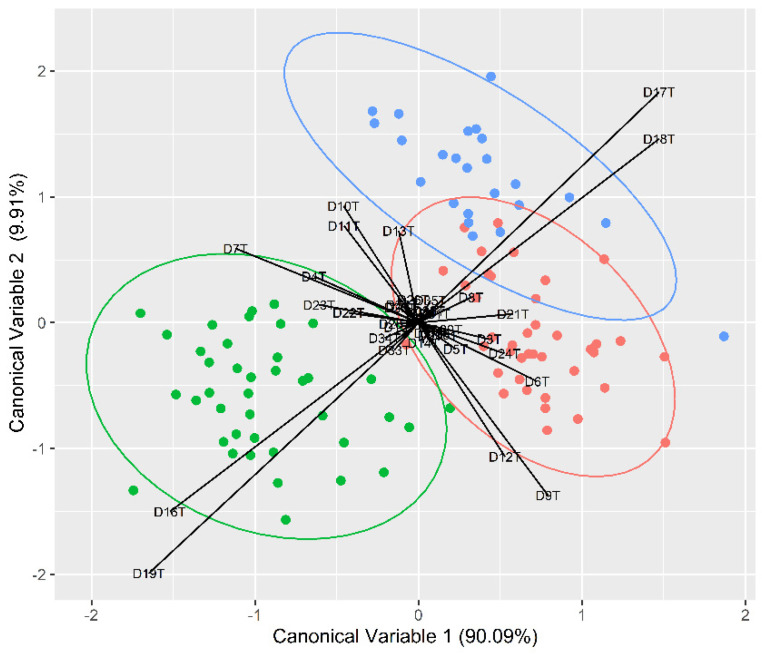
Flexible discriminant function analysis plot obtained from body morphometric transformed distances (the blue, red and green solid circles represent the fish from the eastern Irish Sea, the Cantabria Sea and the northwest Portuguese waters, respectively).

**Table 1 biology-13-00017-t001:** Sampling locations, date, sample size (N) and standard length (SL: mean ± standard deviation) of *Chelidonichthys lucerna* used in this study.

Location	Date	N	SL (cm)
Irish Sea	October 2020	29	19.4 ± 2.9
Cantabria Sea	April 2021	50	28.3 ± 2.0
Portuguese Waters	December 2021	50	22.1 ± 1.7

**Table 2 biology-13-00017-t002:** Body landmarks defined along the body contour of *Chelidonichthys lucerna* and morphometric distances used for the body shape analysis. For more details, please see Figure 2.

Body Landmarks		
Number	Location	
1		Anterior tip of the mouth
2		Anterior margin of the eye
3		Posterior tip of the mouth
4		Anterior insertion of the pelvic fin
5		Anterior insertion of the 1st dorsal fin
6		Posterior insertion of the 1st dorsal fin
7		Anterior insertion of the caudal fin
8		Anterior insertion of the 2nd dorsal fin
9		Posterior insertion of the 2nd dorsal fin
10		Posterior insertion of the caudal fin
11		Ventral insertion of the caudal fin
12		Dorsal insertion of the caudal fin
13		Posterior margin of the caudal peduncle
14R		Right dorsal margin of the head (centre of the eye)
14L		Left dorsal margin of the head (centre of the eye)
15R		Insertion of the right pectoral fin
15L		Insertion of the left pectoral fin
**Morphometric Distances**		
**Distances**	**Landmarks**	**Description**
D1	1 to 2	Head length
D2	1 to 3	Maxilla length
D3	2 to 3	Anterior height of head
D4	2 to 4	Posterior height of head
D5	2 to 5	Distance from the most posterior aspect of neurocranian to the 1st dorsal fin
D6	3 to 4	Distance from maxilla to pelvic fin
D7	3 to 5	Distance from the posterior tip of the mouth to the anterior insertion of the 1st dorsal fin
D8	4 to 5	Anterior body height
D9	4 to 6	Distance from the anterior insertion of the pelvic fin to the posterior insertion of the 1st dorsal fin
D10	4 to 7	Distance between pelvic fin and anal fin
D11	5 to 6	Length of 1st dorsal din
D12	5 to 7	Distance between the origin of 1st dorsal fin and the origin of anal fin
D13	6 to 7	Posterior body height
D14	6 to 8	Distance between 1st and 2nd dorsal fins
D15	7 to 8	Distance from the anterior insertion of the caudal fin to the anterior insertion of the 2nd dorsal fin
D16	7 to 9	Distance from the anterior insertion of the caudal fin to the posterior insertion of the 2nd dorsal fin
D17	7 to 10	Length of anal fin
D18	8 to 9	Length of 2nd dorsal fin
D19	8 to 10	Anterior diagonal height of posterior body
D20	9 to 10	Anterior caudal peduncle height
D21	9 to 11	Anterior diagonal of caudal peduncle
D22	9 to 12	Distance between 2nd dorsal fin and caudal fin
D23	10 to 11	Distance between anal fin and caudal fin
D24	10 to 12	Posterior diagonal of caudal peduncle
D25	11 to 12	Posterior caudal peduncle height
D26	11 to 13	Distance between the ventral insertion of caudal fin and the posterior end of vertebrate column
D27	12 to 13	Distance between the dorsal insertion of caudal fin and the posterior end of vertebrate column
D28	1 to 14R	Distance from the anterior tip of the mouth to the right dorsal margin of the head
D29	1 to 14L	Distance from the anterior tip of the mouth to the left dorsal margin of the head
D30	14R to 14L	Distance between the right and left dorsal margins of the head
D31	14R to 15R	Distance from the right dorsal margin of the head to the insertion of the right pectoral fin
D32	14L to 15L	Distance from the left dorsal margin of the head to the insertion of the left pectoral fin
D33	14R to 15L	Distance from the right dorsal margin of the head to the insertion of the left pectoral fin
D34	14L to 15R	Distance from the left dorsal margin of the head to the insertion of the right pectoral fin
D35	15R to 15L	Distance between the right and left insertions of the pectoral fins
D36	15R to 13	Distance from the insertion of the right pectoral fin to the posterior margin of the caudal peduncle
D37	15L to 13	Distance from the insertion of the left pectoral fin to the posterior margin of the caudal peduncle

**Table 3 biology-13-00017-t003:** Body morphometric transformed distances (DT: mean *±* standard error) calculated for *Chelidonichthys lucerna* individuals. DTs, showing different letters, means that significant regional differences exist. For most DTs, One-Way ANOVA On Ranks, followed by a Dunn test (*p* < 0.05), if needed, was carried out. However, for DT2, DT6, DT8, DT9, DT10, DT11, DT32 and DT35 a One-Way ANOVA and a post hoc pairwise Tukey test were used. For more details, see M&M.

Distance	Irish Sea		Cantabria Sea		Portuguese Waters	
DT1	3.820 ± 0.068	^a^	3.573 ± 0.028	^b^	3.910 ± 0.041	^a^
DT2	2.545 ± 0.068	^a^	2.547 ± 0.035	^a^	2.633 ± 0.035	^a^
DT3	3.820 ± 0.068	^a^	3.573 ± 0.028	^b^	3.910 ± 0.041	^a^
DT4	4.265 ± 0.079	^a^	4.148 ± 0.038	^a^	4.645 ± 0.059	^b^
DT5	4.294 ± 0.081	^a^	4.128 ± 0.024	^a^	4.575 ± 0.058	^b^
DT6	2.668 ± 0.111	^a^	2.522 ± 0.068	^a^	2.991 ± 0.058	^b^
DT7	6.228 ± 0.126	^a^	5.843 ± 0.037	^b^	6.496 ± 0.077	^a^
DT8	4.799 ± 0.075	^a^	4.544 ± 0.043	^b^	4.944 ± 0.062	^a^
DT9	7.175 ± 0.119	^a^	6.830 ± 0.084	^b^	7.360 ± 0.084	^a^
DT10	8.036 ± 0.143	^a^	7.311 ± 0.085	^b^	7.429 ± 0.111	^b^
DT11	3.776 ± 0.069	^a^	3.738 ± 0.059	^a^	4.088 ± 0.051	^b^
DT12	7.081 ± 0.110	^a^	6.628 ± 0.050	^b^	7.142 ± 0.087	^a^
DT13	4.291 ± 0.090	^a^	3.963 ± 0.041	^b^	4.492 ± 0.057	^a^
DT14	1.401 ± 0.093	^a^	1.194 ± 0.046	^a,b^	1.139 ± 0.048	^b^
DT15	3.721 ± 0.072	^a^	3.528 ± 0.033	^b^	4.147 ± 0.049	^c^
DT16	7.632 ± 0.197	^a^	7.226 ± 0.053	^b^	8.020 ± 0.110	^a^
DT17	7.402 ± 0.210	^a^	6.818 ± 0.057	^b^	7.627 ± 0.111	^a^
DT18	7.970 ± 0.162	^a^	7.399 ± 0.059	^b^	7.957 ± 0.115	^a^
DT19	8.372 ± 0.178	^a^	7.669 ± 0.057	^b^	8.415 ± 0.110	^a^
DT20	1.433 ± 0.041	^a^	1.433 ± 0.014	^a^	1.658 ± 0.020	^b^
DT21	1.854 ± 0.077	^a^	1.895 ± 0.023	^b^	2.197 ± 0.040	^a^
DT22	1.368 ± 0.106	^a^	1.416 ± 0.028	^a^	1.739 ± 0.048	^b^
DT23	1.114 ± 0.090	^a^	1.441 ± 0.035	^b^	1.730 ± 0.050	^c^
DT24	1.759 ± 0.060	^a^	1.966 ± 0.028	^b^	2.212 ± 0.043	^c^
DT25	1.198 ± 0.023	^a^	1.115 ± 0.009	^b^	1.159 ± 0.015	^a,b^
DT26	1.484 ± 0.033	^a^	1.286 ± 0.017	^b^	1.133 ± 0.026	^c^
DT27	1.365 ± 0.033	^a^	1.209 ± 0.017	^b^	1.095 ± 0.025	^c^
DT28	3.437 ± 0.084	^a^	3.227 ± 0.025	^b^	3.426 ± 0.042	^a^
DT29	3.482 ± 0.086	^a^	3.297 ± 0.026	^b^	3.506 ± 0.037	^a^
DT30	3.687 ± 0.084	^a^	3.424 ± 0.028	^b^	3.771 ± 0.047	^a^
DT31	2.920 ± 0.093	^a^	2.690 ± 0.042	^b^	3.206 ± 0.046	^c^
DT32	2.786 ± 0.089	^a^	2.600 ± 0.039	^a^	3.247 ± 0.055	^b^
DT33	4.845 ± 0.109	^a^	4.552 ± 0.041	^b^	5.248 ± 0.063	^c^
DT34	4.955 ± 0.122	^a^	4.560 ± 0.042	^b^	5.207 ± 0.054	^c^
DT35	4.313 ± 0.085	^a^	4.008 ± 0.035	^b^	4.473 ± 0.053	^a^
DT36	16.872 ± 0.218	^a^	15.893 ± 0.076	^b^	16.592 ± 0.184	^a^
DT37	17.015 ± 0.215	^a^	15.952 ± 0.081	^b^	16.603 ± 0.194	^a^

**Table 4 biology-13-00017-t004:** Mean and pairwise PERMANOVA comparisons for the 37 body morphometric transformed distances among the three *Chelidonichthys lucerna* sampling locations.

PERMANOVA	Df	SSq	R^2^	F	Pr (>F)	
Region	2	0.057	0.228	18.48	0.0001	
Residual	125	0.192	0.772			
Total	127	0.249	1			
**Pairwise PERMANOVA**	**Df**	**SSq**	**F. Model**	**R^2^**	***p*. value**	***p*. adjusted**
Cantabria Sea vs. Irish Sea	1	0.021	13.989	0.154	0.001	0.003
Cantabria Sea vs. Portuguese Waters	1	0.046	38.707	0.285	0.001	0.003
Irish Sea vs. Portuguese Waters	1	0.014	6.805	0.082	0.002	0.006

Abbreviations: Df—degrees of freedom, SSq—sums of squares, R2—partial R2, F—pseudo F statistic, Pr (>F)—*p* value for each term.

**Table 5 biology-13-00017-t005:** Summary of the percentage of correct reclassification using the training base set following a flexible discriminant analysis (FDA) for the body morphometric transformed distances calculated for *Chelidonichthys lucerna* individuals.

Original Location	Predicted Location			% of Correct Re-Allocation	% of Overall Re-Allocation
	Irish Sea	Cantabria Sea	Portuguese Waters		
Irish Sea	23	1	0	96	95
Cantabria Sea	3	37	0	93	
Portuguese Waters	0	1	39	98	

## Data Availability

The data presented in this study are available on request from the corresponding author.

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
