# Peer review of "Chelidonichthys lucerna (Linnaeus, 1758) Population Structure in the Northeast Atlantic Inferred from Landmark-Based Body Morphometry"

_biology, 2023, doi:10.3390/biology13010017_

Round 1

Reviewer 1 Report

Comments and Suggestions for Authors

Comments for review of the manuscript entitled: “Chelidonichthys lucerna population structure in the northeast Atlantic inferred from landmark-based body morphometry“ ”.

Comments and Suggestions for Authors

The topics covered in the manuscript about the spatial distribution and population structure analyzing fish body shape, a morphometry-based method are not a novelty in biological science and are based on known research methods but it seems to me that the purpose of the undertaken research is justified because it is the first study of the tub gurnard in the three fishing area along the northeast Atlantic. This paper is well-written and uses appropriate analytical methods to quantify population structure. This is a valuable contribution to the management of the species.

Line by Line Comments

Title

My suggestion is to insert Linnaeus, 1758, after the name of the species.

Keywords

Good choice of keywords, which are not reported in the title and will improve the soundness of this manuscript.

Introduction

The introduction is well structured and written in detail with all the necessary information. This section considered the commercial value of C. lucerna. In my opinion, authors should also include parameters such as the fish conservation dimension and the economic size of fish in the study to improve the soundness of the study.

Lines 94 – 98  I suggest to provide the data about the prescribed MLS (minimum landing size) for C. lucerna for the EU and northeast Atlantic if they exist.

Materials and methods

The samples were well collected and in good quantity to confirm the results of this research. In general, the methodological approach and the statistical methods used for each lesson presented are valid and correctly applied to the data.

Line 113 Sampling instead of Biological sampling.

Line 124 I think Table 1 is unnecessary considering the amount of data in it. I suggest displaying the data from the table in the main text.

Results

The analyses were carried out properly. Data interpretation are coherent and well-presented. Tables and figures are also well presented and titled.

Discussion

Please, highlited the limitations of this study.

References

The references list is relevant and has mostly recent publications. 

Author Response

Reviewer 1

Comments and Suggestions for Authors

The topics covered in the manuscript about the spatial distribution and population structure analyzing fish body shape, a morphometry-based method are not a novelty in biological science and are based on known research methods but it seems to me that the purpose of the undertaken research is justified because it is the first study of the tub gurnard in the three fishing area along the northeast Atlantic. This paper is well-written and uses appropriate analytical methods to quantify population structure. This is a valuable contribution to the management of the species.R:

R: Thank you for your positive feed-back.

Line by Line Comments

Title

My suggestion is to insert Linnaeus, 1758, after the name of the species.

R: Suggestion accepted. See revised MS, L2

Keywords

Good choice of keywords, which are not reported in the title and will improve the soundness of this manuscript.

R: Thank you for your positive feed-back.

Introduction

The introduction is well structured and written in detail with all the necessary information. This section considered the commercial value of C. lucerna. In my opinion, authors should also include parameters such as the fish conservation dimension and the economic size of fish in the study to improve the soundness of the study.

R: Suggestion accepted. See revised MS, L100-103 and L400-403.

Lines 94 – 98  I suggest to provide the data about the prescribed MLS (minimum landing size) for C. lucerna for the EU and northeast Atlantic if they exist.

R: Suggestion accepted. See revised MS, L99-100.

Materials and methods

The samples were well collected and in good quantity to confirm the results of this research. In general, the methodological approach and the statistical methods used for each lesson presented are valid and correctly applied to the data.

R: Thank you for your positive feed-back.

Line 113 Sampling instead of Biological sampling.

R: Suggestion accepted. See revised MS, L118.

Line 124 I think Table 1 is unnecessary considering the amount of data in it. I suggest displaying the data from the table in the main text.

R: We understand this suggestion, but we prefer to kept it.

Results

The analyses were carried out properly. Data interpretation are coherent and well-presented. Tables and figures are also well presented and titled.

R: Thank you for your positive feed-back.

Discussion

Please, highlighted the limitations of this study.

R: suggestion accepted. See revised MS, L234-23, L286-294, L299-301, L309-311.

References

The references list is relevant and has mostly recent publications. 

R: Thank you for your positive feed-back.

Reviewer 2 Report

Comments and Suggestions for Authors

This is a well-written ms that provides new information regarding the population structure of the tub gurnard, which is becoming a commercially important species that occurs across a broad geographic region. 

Morphometric variation between regions indicates stock structure and potentially has implications for management, warranting the need for more robust genetic assessment of the population. The data the authors provide are valuable to fisheries managers.

A few comments - in the Intro or Discussion, it would be informative to provide additional context on life history (max size, max age, size at maturity) to compare the samples collected herein to what may be present in the population at large.

Author Response

Reviewer 2

This is a well-written ms that provides new information regarding the population structure of the tub gurnard, which is becoming a commercially important species that occurs across a broad geographic region. 

R: Thank you for your positive feed-back

Morphometric variation between regions indicates stock structure and potentially has implications for management, warranting the need for more robust genetic assessment of the population. The data the authors provide are valuable to fisheries managers.

R: Thank you for your positive feed-back.

A few comments - in the Intro or Discussion, it would be informative to provide additional context on life history (max size, max age, size at maturity) to compare the samples collected herein to what may be present in the population at large.

R: See revised MS, L286-294, and L496-499.

Reviewer 3 Report

Comments and Suggestions for Authors

The reviewed manuscript presents the results of morphometric analyses of Chelidonichthys lucerna from the northeastern Atlantic. Ch. lucerne is a species with high potential for commercial fisheries. Therefore, studies characterising/distinguishing the stocks are needed and very useful. Management of commercially exploited fish populations relies on monitoring of stocks, which we need to be able to identify and distinguish between stocks. The authors of this manuscript have carefully mentioned the above arguments in the Introduction chapter and formulated in a well-defined objective.

The study material consisted of fish caught in three locations, namely Conwy Bay, United Kingdom (Irish Sea), Bay of Biscay, Spain (Cantabrian Sea), and Matosinhos, Portugal. The authors used standard procedures for body shape measurement (truss network) and data analysis. The description of the Results is sufficient. The Discussion chapter is interesting, but could be improved. I believe that the authors should consider the weaknesses of this manuscript. Below are the main comments and specific points that may help to improve this manuscript.

Main comments:

The sample size is not large and the date of collection seems to be the biggest weakness of this research. At each site, samples were collected at different times of the year, which, in conjunction with the life cycle (e.g. spawning period), may have a significant impact on body shape. The authors should supplement the manuscript with this aspect. It should be clarified whether the different seasons of sampling could have influenced the differences in body shape. Also, there is no information on the sex of the analysed individuals. Do both sexes have the same shapes? Also during the spawning season?

Specific points:

l. 61: "anthropic pollution" could the pollution be of another source?

l. 99-100: unclear. What oceanographic barriers do you have in mind?

l. 130-131: can you give details of the digital camera? 

l. 197: Table 3 should be improved in terms of letter indices denoting statistically significant differences. Their form, with large spaces, is not attractive. 

l. 208-209: this is strange because DT19, DT16, DT7 and DT18 were not statistically significantly different between the Irish and Portuguese locations.

Author Response

Reviewer 3

The reviewed manuscript presents the results of morphometric analyses of Chelidonichthys lucerna from the northeastern Atlantic. Ch. lucerne is a species with high potential for commercial fisheries. Therefore, studies characterising/distinguishing the stocks are needed and very useful. Management of commercially exploited fish populations relies on monitoring of stocks, which we need to be able to identify and distinguish between stocks. The authors of this manuscript have carefully mentioned the above arguments in the Introduction chapter and formulated in a well-defined objective.

R: Thank you for your positive feed-back.

The study material consisted of fish caught in three locations, namely Conwy Bay, United Kingdom (Irish Sea), Bay of Biscay, Spain (Cantabrian Sea), and Matosinhos, Portugal. The authors used standard procedures for body shape measurement (truss network) and data analysis. The description of the Results is sufficient. The Discussion chapter is interesting, but could be improved. I believe that the authors should consider the weaknesses of this manuscript. Below are the main comments and specific points that may help to improve this manuscript.

R: Thank you for your positive feed-back. Discussion was improved. See comments below.

Main comments:

The sample size is not large and the date of collection seems to be the biggest weakness of this research. At each site, samples were collected at different times of the year, which, in conjunction with the life cycle (e.g. spawning period), may have a significant impact on body shape. The authors should supplement the manuscript with this aspect. It should be clarified whether the different seasons of sampling could have influenced the differences in body shape. Also, there is no information on the sex of the analysed individuals. Do both sexes have the same shapes? Also during the spawning season?

R: All Suggestions accepted.

Regarding the sampling size, we are confident about it. Our dataset (29 to 50 individuals per region) is similar or better compared to similar studies. This issue was already shortly discussed, but improved now. See Revised MS, L299-301.

Regarding the effect of the reproductive season on the body shape, this issue is discussed as suggested. See revised MS, L286-294, and L496-499.

Regarding potential effect of sex on body shape this is now discussed. See Revised MS, L234-236 and L424 – 426.

Specific points:

L.61: "anthropic pollution" could the pollution be of another source?

R: Yes. You have natural or anthropic pollution. The former is much important. So we highlighted this one

L.99-100: unclear. What oceanographic barriers do you have in mind?

R: Currents, eddies, up-welling, etc. Anyway we change in this sentence geographic to physical ones to be more clear. See Revised MS, L104-106.

L.130-131: can you give details of the digital camera? 

R: Suggestion accepted. We used different cameras (it was a collaboration work among Portuguese, Spanish and English researcher). But the protocol was the same. Anyway we clarified this issue.  See Revised MS, L136-138 and L413-416

L.197: Table 3 should be improved in terms of letter indices denoting statistically significant differences. Their form, with large spaces, is not attractive. 

R: This is the usual form to present this data when we use parametric or non-parametric pairwise comparisons [it means One-Way ANOVA followed if needed (p<0.05) by a Tukey test, or as in our case One-Way Anova on ranks followed if needed (p<0.05) by a Dunn test]. And there are lots of papers using similar tables. However, in the proofs stage (if paper is accepted) we could improve it visually to the reader.

L.208-209: this is strange because DT19, DT16, DT7 and DT18 were not statistically significantly different between the Irish and Portuguese locations.

R: Yes (and we confirm it). But you cannot compare univariate tests (table 3), with multivariate statistocs (fig. 3). The vectors for the flexible DFA are clear.